# Biochemical and Enzymatic Analyses to Understand the Accumulation of γ-Aminobutyric Acid in Wheat Grown under Flooding Stress

**Setsuko Komatsu \*** , **Natsuru Nishiyama and Azzahrah Diniyah**

Faculty of Life and Environmental Sciences, Fukui University of Technology, Fukui 910-8505, Japan
\* Correspondence: skomatsu@fukui-ut.ac.jp; Tel.: +81-276-29-2466

**Abstract:** Flooding induces low-oxygen stress, which reduces plant growth. The activity of the γ-aminobutyric acid (GABA) shunt is crucial for plant stress adaptation, in which it acts by changing cytosolic pH, limiting reactive oxygen species production, regulating nitrogen metabolism, and bypassing steps in the tricarboxylic acid cycle. GABA accumulates under osmotic stress as well as flooding stress. To clarify the dynamic roles of GABA accumulation in wheat under flooding stress, biochemical and enzymatic analyses were performed using a plant-derived smoke solution (PDSS), which rescued wheat growth from flooding stress. Alcohol dehydrogenase abundance increased under flooding stress; however, under the same conditions, pyruvic acid content increased only following PDSS application. Glutamic acid content increased under flooding stress, but decreased following the application of PDSS after 2 days of flooding. Glutamate decarboxylase abundance and GABA content increased under flooding stress, and further increased after 1 day of application of PDSS. Succinyl semialdehyde dehydrogenase accumulated after 2 days of flooding. These results suggest that flooding stress increases GABA content along with the increase and decrease of glutamate decarboxylase and succinyl semialdehyde dehydrogenase, respectively. Additionally, PDSS increased GABA content along with the increase of glutamate decarboxylase abundance at the initial stage of application.

**Keywords:** wheat; plant-derived smoke solution; flooding stress; γ-aminobutyric acid





## 1. Introduction

Climate change has led to significant alterations in global precipitation, temperature, and other related conditions [1]. Climate predictions warn that irregular rainfall, higher average temperature, and $CO_2$ produced by human activity may have a considerable negative influence on the yields of many important field crops [2]. Due to high rainfall, irrigation practices, and poor soil drainage, waterlogging affects large areas of farmland worldwide, resulting in anoxic soils and severe anoxia/hypoxia in crop roots [3]. Low oxygen availability during flooding stress results in enormous yield losses in variety of crops [4]. Wheat plays a crucial role in the economic stability of many countries and is a major cultivated crop all around the world [5]. The tolerance mechanism involved during wheat growth under flooding conditions has not been elucidated, because this tolerance is different among wheat cultivars.

Among abiotic stresses, salt stress inhibited key metabolic enzymes required for the cyclic operation of the tricarboxylic acid (TCA) cycle, which was overcome by increased γ-aminobutyric acid (GABA) shunt activity in wheat [6]. Wheat cultivars responded to drought stress during the seedling stage, which was connected with reactive oxygen species (ROS) scavenging systems and antioxidant enzymes associated with the activation of the GABA shunt pathway and the production of GABA [7]. GABA and glutamic acid had the highest values of centrality in a metabolic correlation network, which indicates their critical roles in the genotype-specific response to nitrogen starvation of wheat [8]. Additionally, in soybean, flooding resulted in a marked decrease of asparagine and a concomitant

accumulation of GABA [9]. Furthermore, the levels of GABA, phosphoenol pyruvate, glycine, and $NADH_2$ increased under flooding stress [10,11]. These results indicate that GABA accumulates not only under osmotic stress but also under flooding stress, suggesting that the accumulation of GABA plays a storage role during flooding stress.

The GABA shunt is a pathway that bypasses the 2-oxoglutarate dehydrogenase complex catalytic step of the TCA cycle [12]. In the GABA shunt pathway, glutamic acid generated from 2-oxoglutarate is decarboxylated to GABA by glutamate decarboxylase (GAD). GABA is catabolized by GABA transaminase (GABA-T) to form succinyl semialdehyde (SSA). The final step is the conversion of SSA to succinate by SSA dehydrogenase (SSADH) [13]. Furthermore, the activity of the GABA shunt is important in adaptation to stresses such as salinity [14], drought [7], and temperature [15] in many kinds of plants. This adaptation occurs by changing the cytosolic pH, limiting ROS production, regulating nitrogen metabolism, and bypassing steps in the TCA cycle [11]. These findings indicate that GABA is involved in the regulation of physiological and biochemical responses in plant cells, including growth and development, signal transmission, and carbon/ nitrogen balance, as well as stress responses.

Plant-derived smoke solution (PDSS) is a substance for promoting seed germination [16,17] and plant growth [18], affecting plant species in various habitats [19]. It positively affects the post-germination growth of wheat [20–22] rice [23–26], maize [27,28], chickpea [29], and soybean [30–33]. PDSS was shown to have the ability to mitigate the phytotoxic effects of heavy-metal stress in wheat by modulating the antioxidative defense system [22]. Furthermore, PDSS enhanced soybean growth under flooding stress [31] and after flooding [30]. Under flooding stress, proteins associated with the ubiquitin–proteasome pathway were altered, leading to the sacrificial survival mechanism of degradation of soybean root tip by PDSS, enabling metabolite accumulation and ensuring lateral root development [31]. During recovery from flooding, PDSS promoted soybean growth through the balance of sucrose/starch metabolism and glycolysis as well as the accumulation of cell-wall-associated proteins [30]. These findings indicate that PDSS is an important substance for stress adaptation in plants.

In the case of wheat, PDSS treatment improved shoot length under normal conditions [20]. The application of PDSS improved the recovery of wheat growth through the regulation of photosynthesis and glycolysis even under flooding conditions, and it promoted tolerance to flooding stress through the regulation of amino acid metabolism [21]. These results indicate that PDSS could help wheat recover from flooding stress. In this study, to reveal the dynamic roles of GABA accumulation in wheat under flooding stress, biochemical and enzymatic analyses were conducted. Furthermore, the alteration of GABA under flooding stress was analyzed using PDSS, which is one of the substances involved in flooding stress adaptation in wheat.

## 2. Materials and Methods

### 2.1. Plant Material and Treatment

Wheat (*Triticum aestivum* L. cultivar Nourin 61) seeds were sterilized with 2% sodium hypochlorite solution, washed, and sown in seedling cases containing 400 mL of silica sand. A total of 20 seeds were sown evenly in each seedling case. Plants were grown in a growth chamber with white fluorescent light (14 h light of 200 $\mu$mol $m^{-2}$ $s^{-1}$ and 10 h dark photoperiod) with 60% humidity at 25 °C. After 3 days, plants were flooded with or without 2000 ppm PDSS (Kohat University of Science and Technology, Kohat, Pakistan) for 1, 2, or 3 days. PDSS was prepared according to previously reported methods [31] (Table S1). Leaf and root samples were collected at 1, 2, and 3 days after flooding. The sowing of seeds was carried out on different days to establish biological replicates. Three independent experiments were performed as biological replicates for all experiments (Figure 1).

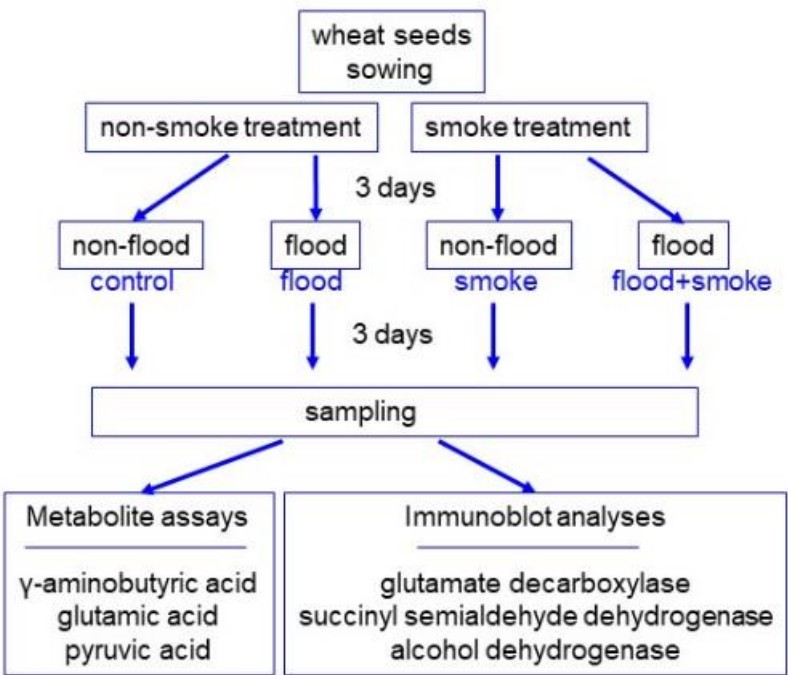

**Figure 1.** Experimental design for investigation of the effect of flooding stress on wheat. To investigate the potential positive or negative effects of flooding stress on wheat, seeds were sown and grown with or without flooding. Three days after sowing, wheat seedlings were flooded for 3 days. Additionally, because PDSS enhances wheat growth under flooding, wheat seedlings were treated with 2000 ppm PDSS. Wheat seedlings were analyzed using metabolite assays and immunoblot analyses. All experiments were performed with three independent biological replicates.

## 2.2. Protein Extraction

Portions (300 mg) of samples were cut into small pieces and ground in 500 μL of lysis buffer, containing 7 M urea, 2 M thiourea, 5% CHAPS, and 2 mM tributylphosphine, in a mortar and pestle. The suspension was centrifuged twice at $16,000 \times g$ for 10 min at 4 °C. Protein concentration was determined via the Bradford method [34].

## 2.3. Immunoblot Analysis

Extracted proteins were added into an SDS sample buffer consisting of 60 mM Tris-HCl (pH 6.8), 2% SDS, 50 mM dithiothreitol, and 10% glycerol as final concentrations [35]. Proteins (10 μg) were separated by electrophoresis on 10% SDS polyacrylamide gel and transferred to polyvinylidene difluoride membranes using a semi-dry transfer blotter (Nippon Eido, Tokyo, Japan). Blotted membranes were blocked with Bullet Blocking One reagent (Nacalai Tesque, Kyoto, Japan) for 5 min. After blocking, the membranes were cross-reacted with the primary antibodies for 30 min. The following primary antibodies were used: anti-SSADH (Abcam, Cambridge, UK), GAD (Bioworld Technology, St. Louis Park, MN, USA) and alcohol dehydrogenase (ADH) [36] antibodies. After reaction, the membrane was cross-reacted using anti-rabbit IgG conjugated with horseradish peroxidase (Bio-Rad, Hercules, CA, USA) as the secondary antibody for 30 min. Signals were detected using the TMB Membrane Peroxidase Substrate kit (Seracare, Milford, MA, USA). Coomassie brilliant blue staining was used as a loading control. Image J software (version 1.53e with Java 1.8.0_172; National Institutes of Health, Bethesda, MD, USA) was used to calculate the integrated densities of the bands.

### 2.4. Assays of Pyruvic Acid, GABA, and Glutamic Acid

Contents of pyruvic acid were analyzed using the Amplite Colorimetric Pyruvate Assay Kit (AAT Bioquest, Sunnyvale, CA, USA). Pyruvate standards, blank controls, and samples were prepared with 50 μL volumes according to the instructions provided by the company. Working solution (50 μL) was added to tubes of pyruvate standard, blank control, and sample to give a total pyruvate assay volume of 100 μL. The reaction mixture was incubated at room temperature for 30 min to 1 h. The absorbance increase was monitored at 575 nm.

The GABA enzymatic assay kit (Enzyme Sensor, Tsukuba, Japan) consisted of two solutions: solution I, containing 10 U/mL ascorbate oxidase, 0.8 U/mL glutamate oxidase, 1200 U/mL catalase, 10 U/mL peroxidase, and 0.8 mM 4-aminoantipyrine; and solution II, containing 2 U/mL GABA-T, 0.8 mM N-ethyl-N-(2-hydroxy-3-sulfopropyl)-3-methylaniline sodium salt, 1 mM sodium 2-oxoglutarate, 2 mM pyridoxal phosphate, and 0.09% sodium azide. For the reaction, 0.5 mL of solution I was added to 50 μL of sample and incubated at 30 °C for 10 min. After incubation, 0.5 mL of solution II was added and incubated at 30 °C for 10 min. After additional incubation, the absorbance was measured at 555 nm. GABA content was determined with reference to the standard curve [37].

The glutamic acid enzymatic assay kit (Enzyme Sensor) also consisted of two solutions, which were GLU-A and GLU-B. The procedure was the same as the method described above.

### 2.5. Statistical Analysis

Data were analyzed using one-way ANOVA followed by Tukey's multiple comparison among multiple groups using SPSS (IBM, Chicago, IL, USA). A $p$-value of less than 0.05 was considered statistically significant.

## 3. Results

### 3.1. Pyruvic Acid Contents and ADH Accumulation in Wheat Treated with PDSS under Flooding Stress

To confirm the effect of PDSS on the anaerobic metabolic system under flooding stress, ADH accumulation and pyruvic acid contents were analyzed. Wheat was treated with PDSS under flooding stress for 1, 2, or 3 days. The abundance of ADH was selectively analyzed using immunoblot analysis (Figure 2). Proteins extracted from roots and leaves were separated on SDS polyacrylamide gel using electrophoresis and transferred onto a membrane. The membrane was cross-reacted with anti-ADH antibody. A staining pattern with Coomassie brilliant blue was used as a loading control (Figure S1). The integrated densities of bands were calculated using ImageJ software with the triplicated immunoblot results. The abundance of ADH significantly increased in the roots both with and without PDSS under flooding stress after 1 day of treatment. It also increased in the leaves subjected to flooding stress for 2 days; however, it did not increase with PDSS even under flooding stress (Figure 2). For the contents of pyruvic acid, the time dependency effects of treatments were analyzed. The content of pyruvic acid significantly increased in the leaves treated with PDSS under flooding stress after 2 days of treatment. It also significantly increased in roots treated with PDSS under flooding stress for 3 days; however, it did not change in the absence of PDSS even under flooding stress (Figure 3).

### 3.2. Immunoblot Analysis of Proteins Related to GABA Synthesis and Degradation in Wheat Treated with PDSS under Flooding Stress

As proteins related to GABA synthesis and degradation in wheat under flooding stress, the abundances of GAD and SSADH were selectively analyzed using immunoblot analysis (Figures 4 and 5). Proteins extracted from roots and leaves were separated on SDS polyacrylamide gel using electrophoresis and transferred to membranes. The membranes were cross-reacted with anti-GAD and SSADH antibodies. Coomassie brilliant blue staining pattern was used as a loading control (Figure S1). The integrated densities of bands were calculated from the triplicated immunoblot results using ImageJ software. The abundance

of GAD increased under 2 and 3 days of flooding stress; it further increased in roots treated with PDSS under 1 day of flooding stress. After 3 days of flooding stress, GAD levels increased with PDSS treatment. In wheat leaves, the abundance of GAD increased only with PDSS treatment while under flooding stress (Figure 4). The abundance of SSADH increased only under 1 day of flooding stress treatment in leaves and after 2 days of flooding stress treatment in roots. These amounts decreased in both leaves and roots following 3 days of flooding stress treatment (Figure 5).

### 3.3. Assays of GABA and Glutamic Acid Contents in Wheat Treated with PDSS under Flooding Stress

The content of glutamic acid significantly increased in roots treated with and without PDSS under flooding stress in the 1-day treatment group. It decreased in roots treated with PDSS under flooding stress in the 2-day treatment group; however, it continuously increased under flooding stress without PDSS treatment (Figure 6). The content of GABA significantly increased in roots under flooding stress in the 1-day treatment group; it further increased when seedlings were treated with PDSS under flooding stress. However, it decreased in roots treated with PDSS under flooding stress in the 2-day treatment group (Figure 7).

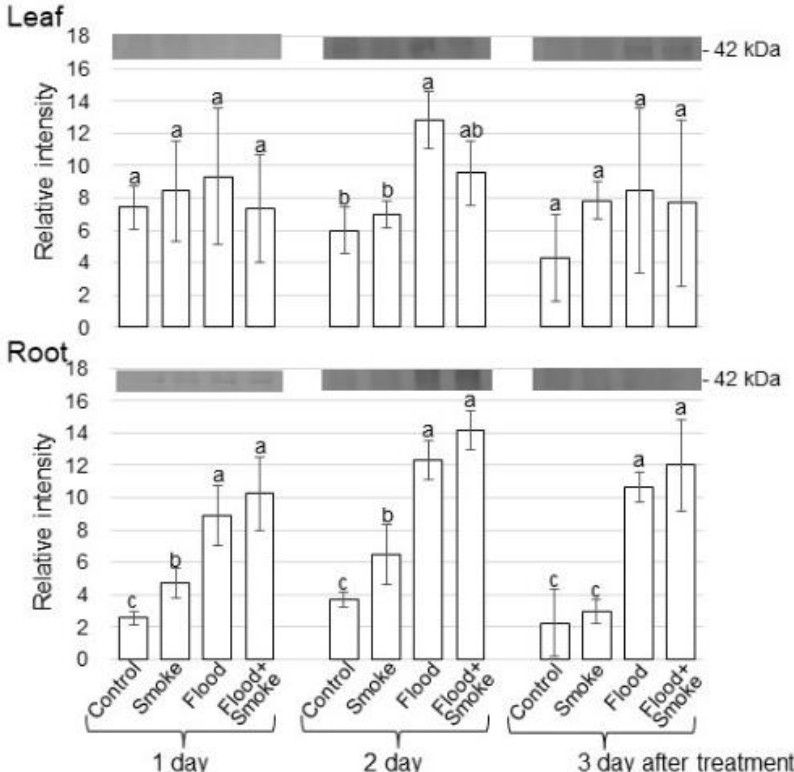

**Figure 2.** Immunoblot analysis of proteins involved in alcohol fermentation in wheat treated with PDSS under flooding stress. Proteins extracted from leaf and root samples of wheat seedlings were separated on SDS polyacrylamide gel using electrophoresis and transferred to membranes. The membranes were cross-reacted with anti-ADH antibody. Coomassie brilliant blue staining pattern was used as a loading control (Figure S1). ImageJ software was used to calculate the integrated densities of the bands. The data are presented as mean ± SD from three independent biological replicates (Figure S2). Mean values with different letters are significantly different according to one-way ANOVA followed by Tukey's multiple comparisons ($p < 0.05$).

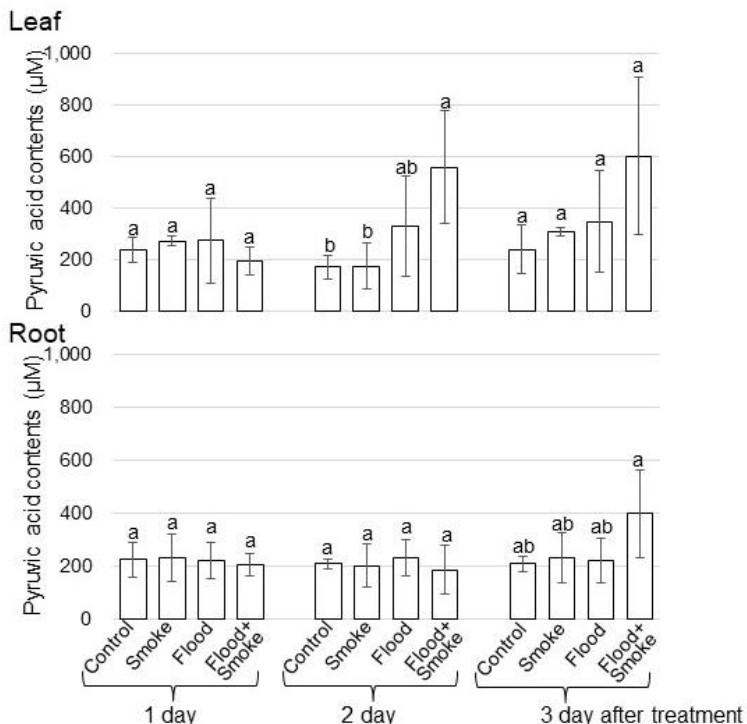

**Figure 3.** Contents of pyruvic acid in wheat treated with PDSS under flooding stress. Samples extracted from leaf and root samples of wheat seedlings were analyzed in terms of the content of pyruvic acid. The data are presented as mean $\pm$ SD from three independent biological replicates. Mean values with different letters are significantly different according to one-way ANOVA followed by Tukey's multiple comparisons ($p < 0.05$).

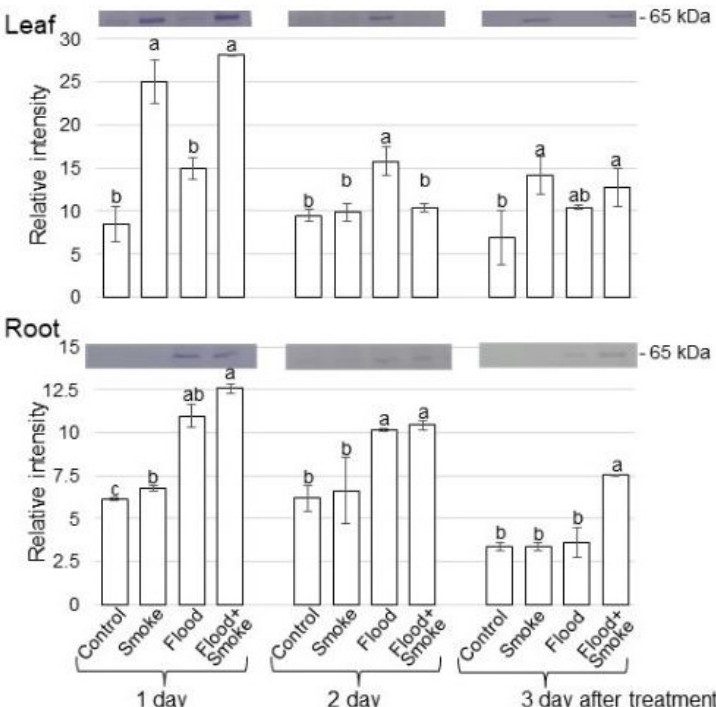

**Figure 4.** Immunoblot analysis of GAD in wheat seedlings treated with PDSS while under flooding stress. Proteins extracted from leaves and roots of wheat seedlings were separated on SDS polyacrylamide gel using electrophoresis and transferred onto membranes. The membranes were cross-reacted with anti-GAD antibody. Coomassie brilliant blue staining pattern was used as a loading control (Figure S1).

ImageJ software was used to calculate the integrated densities of the bands. The data are presented as mean ± SD from three independent biological replicates. Mean values with different letters are significantly different according to one-way ANOVA followed by Tukey's multiple comparisons ($p < 0.05$).

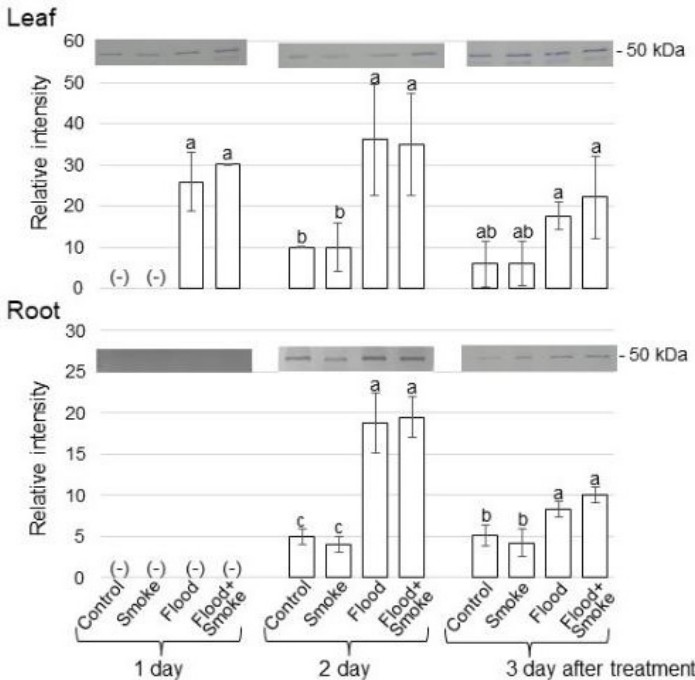

**Figure 5.** Immunoblot analysis of SSADH in wheat treated with PDSS under flooding stress. Proteins extracted from leaf and root samples of wheat seedlings were separated on SDS polyacrylamide gel using electrophoresis and transferred onto membranes. The membranes were cross-reacted with anti-SSADH antibody. Coomassie brilliant blue staining pattern was used as a loading control (Figure S1). ImageJ software was used to calculate the integrated densities of the bands. The data are presented as mean ± SD from three independent biological replicates. Mean values with different letters are significantly different according to one-way ANOVA followed by Tukey's multiple comparisons ($p < 0.05$).

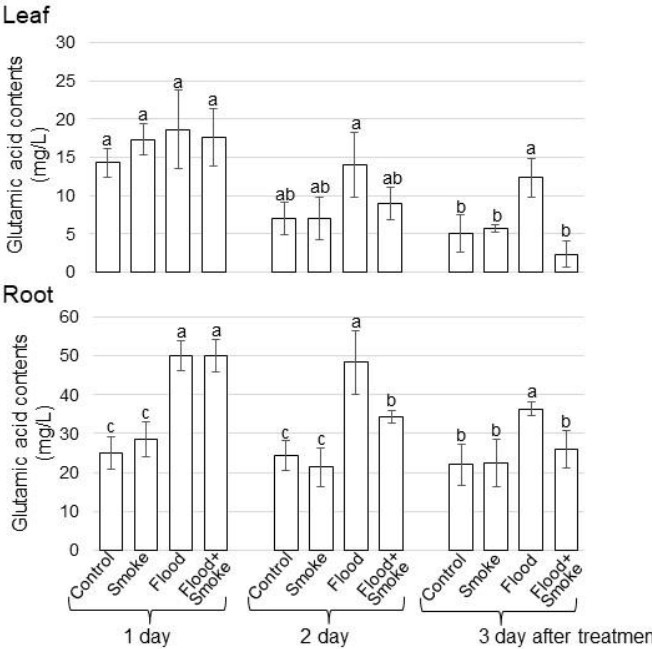

**Figure 6.** Contents of glutamic acid in wheat seedlings treated with PDSS under flooding stress. Samples extracted from leaves and roots of wheat seedlings were analyzed in terms of the content of

glutamic acid. The data are presented as mean ± SD from three independent biological replicates. Mean values with different letters are significantly different according to one-way ANOVA followed by Tukey's multiple comparisons ($p < 0.05$).

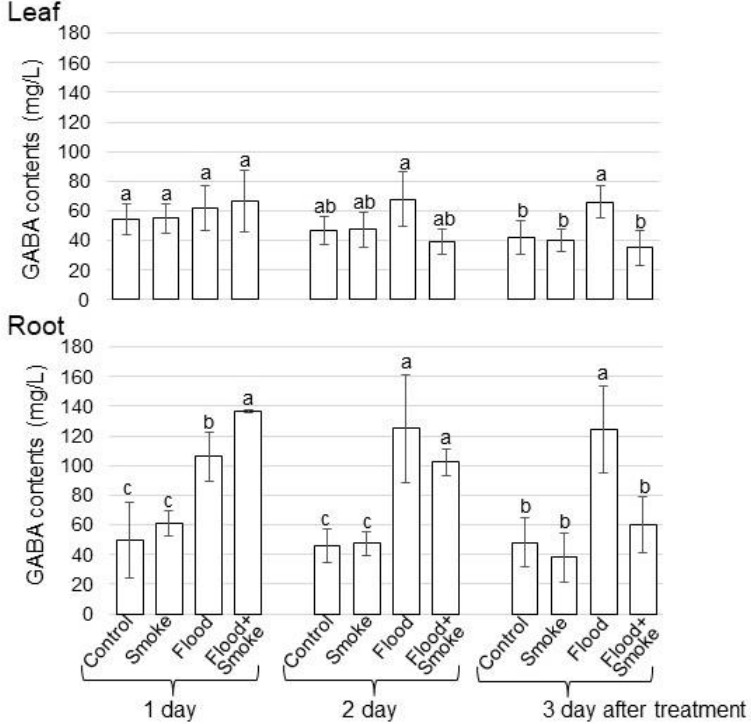

**Figure 7.** GABA contents in wheat seedlings treated with PDSS under flooding stress. Samples extracted from leaves and roots of wheat seedlings were analyzed in terms of GABA content. The data are presented as mean ± SD from three independent biological replicates. Mean values with different letters are significantly different according to one-way ANOVA followed by Tukey's multiple comparisons ($p < 0.05$).

## 4. Discussion

To confirm the effects of flooding stress on anaerobic metabolic systems, ADH accumulation and pyruvic acid contents were analyzed. ADH increased in roots under flooding stress, but it was not significantly changed in leaves (Figure 2). In the aerobic pathway, pyruvic acid is converted into acetyl-coenzyme A and drives ATP production via oxidative phosphorylation and the TCA cycle [38]. Since the wheat leaves were close to the surface of the water, they may have been able to use oxygen from the air to avoid stress. Additionally, pyruvic acid contents did not change in the roots and leaves after 3 days of flooding stress (Figure 3). In Potamogetonaceae, the activation of glycolysis and fermentation processes was recorded and showed similar increases in ADH activity and pyruvic acid concentration after being exposed to low-oxygen conditions [39]. Pyruvate decarboxylase catalyzes the first step in the alcohol fermentation pathway, which is responsible for the irreversible conversion of pyruvic acid to acetaldehyde; additionally, it is the source of metabolic regulation in this pathway [40]. Attempts made to alter the levels of pyruvate decarboxylase and alcohol dehydrogenase in transgenic plants did not yield significantly enhanced levels of low-oxygen stress tolerance [4]. The switch from respiration to fermentation is not completely understood, although the importance of the fermentation pathway in hypoxia survival is known.

Using metabolomic analysis, the levels of GABA, glycine, NADH₂, and phosphoenol pyruvate were identified as having increased under flooding stress [10,11]. To reveal the roles of GABA accumulation in wheat under flooding stress, biochemical and enzymatic

analyses were conducted in a time-dependent manner. Under flooding stress, glutamic acid content increased (Figure 6), as did GAD abundance and GABA content (Figures 4 and 7). SSADH abundance also increased after 2 days of flooding stress (Figure 5). The changes in GABA content were closely related to the enzyme activity and transcription levels of three key enzymes, which are GAD, GABA-T, and SSADH in the GABA shunt [41]. The accumulation of endogenous GABA by the GAD enzyme prevents cytosolic acidification during stress and senescence [42]. These results, along with previous findings, suggest that flooding stress increases GABA content along with the increase and decrease of GAD and SSADH, respectively.

Cell survival is related to energy metabolism, within which oxidative phosphorylation and glycolysis are the main ATP biosynthesis pathways [43]. Pyruvic acid, which is the key end-product of glycolysis, plays important roles in regulating energy metabolism since it is located at the crossroads of the aerobic and anaerobic pathways [44]. In the anaerobic pathway, pyruvic acid is converted into lactate under the catalysis of lactate dehydrogenase and the coenzyme NADH to provide energy in tumor cells [45]. The production of pyruvic acid can be reduced via alanine aminotransferase, which catalyzes the reversible reaction interconverting pyruvic acid and glutamic acid to alanine and 2-oxoglutarate [46]. Additionally, oxoglutarate has been indicated to reenter the TCA cycle to be used to produce another ATP and succinate, which accumulates in the cell [46]. The phosphorylation levels of differentially accumulated phosphoproteins, which are involved in pyruvic acid metabolism and energy production, were identified in response to flooding in *Kandelia candel* [47]. Exogenous GABA enhanced tolerance to hypoxia in melon and storage performance of postharvest citrus fruits via the accumulation of endogenous amino acids and the promotion of the TCA cycle [48]. Moreover, as a key signaling molecule, GABA is involved in barley [49] and soybean [50] responses to stresses such as salt. These findings indicate that the GABA shunt might contribute to stress tolerance, including hypoxic stress. In this study, PDSS was used as a method of indicating tolerance to flooding stress.

The tolerance of wheat to waterlogging depends on the plant's ability to change its morphological and metabolic traits in response to stress to ensure its survival and growth [51]. Therefore, the tolerant plant genotypes can adapt to waterlogging or submergence by developing different metabolic traits to at least survive under hypoxic or anoxic stress [52]. Wheat growth was improved by the application of PDSS through the regulation of photosynthesis, glycolysis, and amino acid metabolism, even under flooding conditions [21]. The present results indicate that the improvement of wheat growth through the application of PDSS requires the increase of pyruvic acid content (Figure 3). GABA contents increased under flooding and further increased with the application of PDSS in wheat [21]. In this study, glutamic acid content increased under flooding stress; however, it decreased with the application of PDSS after a 2-day flooding treatment (Figure 6). GAD abundance and GABA content increased under flooding stress and further increased after 1 day of application of plant-derived smoke (Figures 4 and 7). SSADH abundance increased after 2 days of flooding stress with or without the application of PDSS (Figure 5). *Cannabis sativa* smoke may be able to interact with GABA and glycine receptors [53]. These results, along with previous findings, suggest that the application of PDSS increases GABA content along with increased GAD abundance at the initial stage of application.

A number of reverse-genetic experiments have revealed positive relationships between GABA levels and tolerance to stresses [54]. Furthermore, the application of exogenous GABA reduced ROS levels, enhanced membrane stability, modulated phytohormone cross-talk, and improved tolerance against multiple stresses [55]. In this study, hypoxic flooding stress induced the accumulation of GABA in wheat roots and leaves. It is possible that GABA regulates cytoplasmic pH, sustains nitrogen/carbon fluxes and the TCA cycle, or increases antioxidant stress tolerance to improve wheat tolerance to flooding stress [56–58]. GABA catabolism prevents ROS generation and suppresses cell death, indicating that GABA reduces ROS damage to enhance plant seedling survival during flooding.

## 5. Conclusions

Wheat plays a crucial role in the economic stability of many countries and is a major cultivated field crop all around the world [5]. However, its growth is hampered by flooding, which induces low-oxygen stress. Recently, it was reported that GABA content significantly increased under flooding stress [21]. The activity of the GABA shunt is important for stress adaptation in plants, as it is involved in changing cytosolic pH, limiting ROS production, regulating nitrogen metabolism, and bypassing steps in the TCA cycle [11]. To reveal the dynamic roles of GABA accumulation in wheat under flooding stress, biochemical and enzymatic analyses were conducted using PDSS, which saves wheat from flooding stress. The main findings were as follows: (1) ADH abundance increased under flooding stress, but pyruvic acid content increased only with PDSS under the same conditions; (2) glutamic acid content increased under flooding stress but decreased with application of PDSS after 2 days of flooding treatment; (3) GAD abundance and GABA content increased under flooding stress and further increased after 1 day of application of plant-derived smoke; and (4) SSADH abundance increased after 2 days of flooding stress. These results suggest that flooding stress increases GABA content with the increase and decrease of GAD and SSADH, respectively. Additionally, PDSS increased GABA content with the increase of GAD abundance at the initial stage of application. Furthermore, these findings are useful for future research exploring the molecular mechanisms of GABA-mediated improvements in crop productivity under low-oxygen stress conditions.

**Supplementary Materials:** The following supporting information can be downloaded at: https://www.mdpi.com/article/10.3390/oxygen3010009/s1. Table S1. Preparation of plant-derived smoke solution. Figure S1.; The Coomassie brilliant blue staining patterns of proteins used for immunoblot analysis.; Figure S2. Blots of the entire membrane with ADH antibody, which were used in the preparation of Figure 2.

**Author Contributions:** Conceptualization, S.K.; sample preparation, N.N., A.D. and S.K.; biological experiments and data analyses, N.N., A.D. and S.K.; writing, review, and editing, S.K.; All authors have read and agreed to the published version of the manuscript.

**Funding:** This work was supported by Research Grants from Fukui University of Technology (FY 2022) to S.K.

**Institutional Review Board Statement:** Not applicable.

**Informed Consent Statement:** Not applicable.

**Data Availability Statement:** The data is contained within the article and Supplementary.

**Acknowledgments:** Authors thank to Shafiq Ur Rehman of Kohat University of Science and Technology for providing with PDSS.

**Conflicts of Interest:** The authors declare no conflict of interest.

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
