# Peer review of "Biochemical and Enzymatic Analyses to Understand the Accumulation of γ-Aminobutyric Acid in Wheat Grown under Flooding Stress"

_oxygen, doi:10.3390/oxygen3010009_

Round 1
Reviewer 1 Report
The paper presents and interesting study on the effect of smoke on wheat under floof stress.
I have only one important concern. Figures 2,4 and 5 are depicted as immunoblot analysis, but in fact are quantifications of bands that appeared in the immunoblot. Without having the immunoblot is difficult to evaluate the accuracy of the figures. Please include the immunoblots in the figure or as supplemental data.
Minor points:
Lines 34-35: Many countries.
Line 111: grounded.
Author Response
Reviewer 1
The paper presents and interesting study on the effect of smoke on wheat under floof stress.
I have only one important concern. Figures 2,4 and 5 are depicted as immunoblot analysis, but in fact are quantifications of bands that appeared in the immunoblot. Without having the immunoblot is difficult to evaluate the accuracy of the figures. Please include the immunoblots in the figure or as supplemental data.
Answer: Thank you very much for your advice. Immunoblot pictures have been added in Figures 2,4, and 5. Additionally, the Coomassie-brilliant blue staining patterns have been added as new Supplemental Figure 1.
Minor points:
Lines 34-35: Many countries.
Answer: Thank you very much for your correction. This error has been corrected. Additionally, the rest of the manuscript has also been carefully corrected.
Line 111: grounded.
Answer: The word “ground” has been used as the past participle of “grind”
Reviewer 2 Report
The manuscript deals with the evaluation of wheat cultivated unde flooding stress and measurement of biochemical and enzymatic traits. Authors have proposed to study the Accu-2 mulation of γ-Aminobutyric Acid.
The manuscript is readible but many shortcomings are present.
1-why authors used plant-derived smoke solution for treatments? This is neither presetend in title nor described as a treatment in different sections.
2-how authors have obtained plant-derived smoke solution? this is lacking.
3- the experimental design is not clear. First it is not described. The figure 1 (entitled "Experimental design for investigation of the effect of plant-derived smoke solution on 102 wheat under flooding stress") did not present this experimental design. Nevertheless, we can guess that it is a split plot design (somke treatement factor and flooding as sub factor). If this is the case, the statistical analyses should be redone accordingly. Please precise the use mean comparison test (Duncan, Tukey
4- why authors, have evaluated the germination ?
5- the conclusion should be modified to give more perspectives.
Minos comments
Author Response
Reviewer 2
The manuscript deals with the evaluation of wheat cultivated unde flooding stress and measurement of biochemical and enzymatic traits. Authors have proposed to study the Accu-2 mulation of γ-Aminobutyric Acid. The manuscript is readible but many shortcomings are present.
Answer: We are sorry for this problem. Based on comments from three reviewers, this article has been improved.
1-why authors used plant-derived smoke solution for treatments? This is neither presetend in title nor described as a treatment in different sections.
Answer: Thank you very much for pointing this out. Based on the comments, the reason for using plant-derived smoke has been added in the section “Introduction” in red. Because the preparation of plant-derived smoke was the same as in a previous publication (Zhong et al., J Proteomics, 2018, 181, 238–248), this publication has been cited in the section “2.1. Plant Material and Treatment” in red. Finally, the effect of plant-derived smoke on GABA accumulation was low, although GABA was clearly accumulated under flooding stress. Therefore, the words “plant-derived smoke” were deleted from the title.
2-how authors have obtained plant-derived smoke solution? this is lacking.
Answer: We are sorry for this problem. Because the preparation of plant-derived smoke was the same in a previous publication (Zhong et al., J Proteomics, 2018, 181, 238–248), this publication has been cited in the section “2.1. Plant Material and Treatment” in red. It has been written as follows: “Three-day-old plants were flooded with or without 2,000 ppm plant-derived smoke solution (Kohat University of Science and Technology, Kohat, Pakistan) for 1, 2, and 3 days [23].” Additionally, a more detailed explanation is provided in Supplemental Table 1.
3- the experimental design is not clear. First it is not described. The figure 1 (entitled "Experimental design for investigation of the effect of plant-derived smoke solution on 102 wheat under flooding stress") did not present this experimental design. Nevertheless, we can guess that it is a split plot design (somke treatement factor and flooding as sub factor). If this is the case, the statistical analyses should be redone accordingly. Please precise the use mean comparison test (Duncan, Tukey
Answer: Thank you very much for correctly understanding the situation; smoke treatment under flooding is indeed a sub factor. The legend of Figure 1 has been corrected. Based on suggestion from reviewer, the statistical analyses have been redone accordingly. Furthermore, some of experiments for immunoblots have been performed, again.
4- why authors, have evaluated the germination ?
Answer: Because germination ratio did not change in the case of wheat, we did not write about it in this manuscript. In this research, wheat seedling after germination was used.
5- the conclusion should be modified to give more perspectives.
Answer: We are sorry. As suggested, the conclusion has been rewritten in red.
Reviewer 3 Report
At first glance, this is a well-structured manuscript. The methodology is well described. In particular, Figure 1 is very helpful in understanding the test procedure.
The authors investigated whether plant-derived smoke can mitigate the negative effects of flood stress on wheat seedlings. To do this, they subjected 3-day-old seedlings to water logging stress for a period of 3 days, and carried out the measurements shown on the following 3 days. The measured values ​​therefore come from seedlings aged 7, 8 and 9 days. - At this stage of development, metabolism in seedlings changes from being completely dependent on substrate import from the grain to being self-sufficient through photosynthetic activity in the developing leaves. Parallel to this development, the importance of mitochondrial and plastidic metabolism for the development of seedlings is changing. This connection can only be adequately assessed by comparing the metabolic activities in the storage tissue of the wheat grain, the developing leaves, and the root. The data presented in the manuscript are therefore incomplete. Nevertheless, a progression from day 7 to day 9 might be discussed for each of the measured parameters. (But the authors refrained from doing so.)
Stress from lack of oxygen actually causes significant losses in wheat crops. Accordingly, there is already an extensive number of publications on this problem as reviewed by Agarwal and Grover in Critical Reviews in Plant Sciences, 25:1–21, 2006, for instance. The authors are mentioning that application of plant-derived smoke is an old technique to improve stress tolerance. As a proof they are citing quite recent (2017) papers along with publication from their team published in 2020, 2021, 2022 (references 13, 15, 25). They are not mentioning earlier publications, such as Kulkami et al. South African J. Bot. 77:972-979 (2011). - These are just two examples of an unbalanced consideration of the current literature.
The team of authors is analyzing adverse effects of water logging for quite a while already. Accordingly, they have contributed to 6 out of 38 publications cited in this manuscript. Individual results from these publications are quoted. These citations are not easily to be checked directly if the journals are not freely available online. For instance, the production of plant-derived smoke is not described and the composition is not mentioned. Rather, reference is made to citation 23, a publication by the team that is not freely available online. The authors therefore should consider providing a short description of the employed method. Otherwise interested readers will not be able to reproduce the presented results.
In their experiments the authors have not only applied flooding but also have analyzed beneficial effects of treatment with plant-derived smoke solution. This additional treatment has been analysed in much detail. The authors might indicate this in the headline of their publication.
In an earlier publication team members have contributed to (Zhong et al. 2020) it is described that plant-derived smoke solution is stimulating in soybeans the ornithin synthesis pathway as well as ubiquitin mediated recycling of biomass to overcome metabolite shortage during spells of stress. - The authors are referring to these results in the introduction as well as the discussion chapters. But they do not comment on the different preference for asparagine and ureides as substrates for the transport of assimilated nitrogen in wheat and soybean, respectively. - They did not try to explain the reason for different observations made in the current manuscript.
All measurements have been performed on shoots as well as roots. Significant differences are shown in some of the figures. But the authors refrain from mentioning this observation in the discussion chapter. Rather, the authors cite the results of other publications in their discussion, while only presenting the measured values ​​from their experiments. Overall, the discussion lacks the development of a concept with which the observations can be explained. - In the present form, this manuscript does not offer any new insights. All observations are already known from the existing literature.
Author Response
Reviewer 3
At first glance, this is a well-structured manuscript. The methodology is well described. In particular, Figure 1 is very helpful in understanding the test procedure.
The authors investigated whether plant-derived smoke can mitigate the negative effects of flood stress on wheat seedlings. To do this, they subjected 3-day-old seedlings to water logging stress for a period of 3 days, and carried out the measurements shown on the following 3 days.
The measured values ​​therefore come from seedlings aged 7, 8 and 9 days. - At this stage of development, metabolism in seedlings changes from being completely dependent on substrate import from the grain to being self-sufficient through photosynthetic activity in the developing leaves. Parallel to this development, the importance of mitochondrial and plastidic metabolism for the development of seedlings is changing. This connection can only be adequately assessed by comparing the metabolic activities in the storage tissue of the wheat grain, the developing leaves, and the root. The data presented in the manuscript are therefore incomplete. Nevertheless, a progression from day 7 to day 9 might be discussed for each of the measured parameters. (But the authors refrained from doing so.)
Answer: We are sorry for this problem. In this research, the growth stages of wheat plant were at 4, 5, and 6 days, which were at 1, 2, and 3 days after flooding stress. In near future, using samples from seedlings aged 7, 8 and 9 days, metabolite contents and enzyme activities will be analyzed. The section “Materials and Methods” with Figure 1 has been modified because experimental procedure was not clear. Additionally, the section “Discussion” has been improved based on comments from reviewers.
Stress from lack of oxygen actually causes significant losses in wheat crops. Accordingly, there is already an extensive number of publications on this problem as reviewed by Agarwal and Grover in Critical Reviews in Plant Sciences, 25:1–21, 2006, for instance. The authors are mentioning that application of plant-derived smoke is an old technique to improve stress tolerance. As a proof they are citing quite recent (2017) papers along with publication from their team published in 2020, 2021, 2022 (references 13, 15, 25). They are not mentioning earlier publications, such as Kulkami et al. South African J. Bot. 77:972-979 (2011). - These are just two examples of an unbalanced consideration of the current literature.
Answer: Thank you very much for your suggestion. We are sorry that we could not find the paper you suggested us. Now “Agarwal and Grover in Critical Reviews in Plant Sciences, 25:1–21, 2006” has been cited in this article. However, we are sorry that we could not read “Kulkami et al. South African J. Bot. 77:972-979 (2011)”. So, other publications have been cited in the section “Introduction” as follows:
*. De Lange, J.H.; Boucher, C. Auto ecological studies on Audinia capitata (Bruniaceaae), plant-derived smoke as a germination cue. S. Afr. J. Bot. 1990, 56, 188–202.
*. Brown, N.A.C. Promotion of germination of fynboss seeds by plant-derived smoke. New Phytol. 1993, 123, 575–583.
*. Van Staden, J.; Sparg, S.G.; Kulkarni, M.G.; Light, M.E. Post-germination effects of the smoke-derived compound 3-methyl-2H-furo[2,3-c] pyran-2-one, and its potential as a preconditioning agent. Field Crops Res. 2006, 98, 98–105.
The team of authors is analyzing adverse effects of water logging for quite a while already. Accordingly, they have contributed to 6 out of 38 publications cited in this manuscript. Individual results from these publications are quoted. These citations are not easily to be checked directly if the journals are not freely available online. For instance, the production of plant-derived smoke is not described and the composition is not mentioned. Rather, reference is made to citation 23, a publication by the team that is not freely available online. The authors therefore should consider providing a short description of the employed method. Otherwise interested readers will not be able to reproduce the presented results.
Answer: We are sorry for this problem. As suggested, the production and composition of plant-derived smoke have been described in new supplemental table 1. Additionally, a short description of the method employed in citation 23 has been added in new supplemental table 1.
In their experiments the authors have not only applied flooding but also have analyzed beneficial effects of treatment with plant-derived smoke solution. This additional treatment has been analysed in much detail. The authors might indicate this in the headline of their publication.
Answer: Thank you very much for your suggestion. Plant-derived smoke treatment under flooding is indeed a sub factor; so, it has been removed from the headline of this article. The legend of Figure 1 has been corrected and many parts have been corrected in red.
In an earlier publication team members have contributed to (Zhong et al. 2020) it is described that plant-derived smoke solution is stimulating in soybeans the ornithin synthesis pathway as well as ubiquitin mediated recycling of biomass to overcome metabolite shortage during spells of stress. - The authors are referring to these results in the introduction as well as the discussion chapters. But they do not comment on the different preference for asparagine and ureides as substrates for the transport of assimilated nitrogen in wheat and soybean, respectively. - They did not try to explain the reason for different observations made in the current manuscript.
Answer: As suggested, the reason for different observations between previous publication and the current manuscript has been explain in the section “discussion” in red. Furthermore, the different preference for asparagine and ureides as substrates for the transport of assimilated nitrogen in wheat and soybean has been commented in the section “discussion” in red.
All measurements have been performed on shoots as well as roots. Significant differences are shown in some of the figures. But the authors refrain from mentioning this observation in the discussion chapter. Rather, the authors cite the results of other publications in their discussion, while only presenting the measured values ​​from their experiments.
Answer: Thank you very much for your suggestion. Differences between shoot and root have been commented. And the results in this research have been discussed in detail as suggested.
Overall, the discussion lacks the development of a concept with which the observations can be explained. - In the present form, this manuscript does not offer any new insights. All observations are already known from the existing literature.
Answer: The section “discussion” has been re-written for the results in this research in red as suggested. Additionally, the conclusion has been modified to give more perspectives.
Round 2
Reviewer 1 Report
Paper has been significatively improved. Can be accepted
Reviewer 2 Report
Authors have addressed all my concerns in this manuscript. this latter is now clearer.
Reviewer 3 Report
I have read the response of the authors that has been sent along with
the revised manuscript. It appears to me that the manuscript now is acceptable.A remaining problem is the significant share of self citations of the
team of authors.